# Research on the influence of the morphology of coarse aggregates from Reclaimed Asphalt Pavement (RAP) on the mechanical properties of asphalt concrete based on finite element modeling

**Nan Ru[1], Yu Zhao[2]\*, Zhaoting Liu[3], Ya Li[1], Shuangchen Xia[4], Wencheng Cheng[4]**

**1** School of Civil Engineering, Chongqing Jiaotong University, Chongqing 400074, China, **2** Chengdu Jiaotou Construction Industrialization Co., Ltd., Chengdu 614100 , China, **3** BYD Company Limited, Shenzhen, Guangdong 518038, China, **4** Sichuan Chengqiongya Expressway Co., Ltd., Chengdu 610000, China

\* dkhs2020@outlook.com

## Abstract

Using recycled coarse aggregates (RCA) to prepare asphalt mixtures can reduce the consumption of natural sand and gravel resources and solve the environmental pollution problem caused by waste asphalt concrete materials. To promote the widespread application of RCA, research has been conducted on fully recycled asphalt concrete (FRCAC). The ABAQUS finite – element software was used, and a random aggregate placement algorithm for RCA was implemented by writing the built – in scripting language Python to generate digital specimens. After obtaining the material parameters through indoor tests, the digital specimens were analyzed. Four aggregate characteristic parameters, namely aspect ratio, shape parameter, roundness ratio, and angularity, were selected to further describe the relationship between the external features of RCA and the mechanical properties of asphalt concrete. The results show that: (1) As the irregularity of the RCA's external shape increases, the ultimate failure load of the specimens shows an upward trend. (2) When the aspect ratio reaches 1.70 to 1.75, the shape parameter reaches 0.70 to 0.75, the roundness ratio is between 0.84 to 0.82, and the angularity reaches 0.76 to 0.74, the ultimate failure load of the specimens shows a downward trend. (3) The finite-element modeling algorithm can well reflect the morphology of coarse aggregates and accurately characterize the influence of aggregate morphology on the mechanical properties of asphalt mixtures.

## 1. Introduction

As one of the important components of asphalt mixture, the structure constructed by the interlocking of aggregates is the main bearing structure of asphalt concrete.

**Data availability statement:** All relevant data are within the paper and its Supporting information files.

**Funding:** The author(s) received no specific funding for this work.

**Competing interests:** The authors declare that they have no known competing financial interests or personal relationships that could have appeared to influence the work reported in this paper.

Traditional research mostly relies on indoor tests (such as uniaxial compression and indirect tension) to evaluate the material properties. However, the test results are easily affected by the discreteness of sample preparation, and it is difficult to separate the independent contribution of the aggregate morphology [1]. Zhang Jingna et al. [2] carried out splitting tests and creep tests on different types of asphalt mixtures and established an equation method for predicting the fatigue performance based on the cumulative flow energy consumption. Du Qunle et al. [3] used fatigue tests under the stress control mode to conduct a comparative analysis of the fatigue performance of specimens prepared by the rotary compaction method and the Marshall compaction method for different mixtures, and studied the influence of the nominal maximum particle size on the fatigue performance of the mixtures.

With the advancement of X-ray computed tomography (X-ray CT) and digital image processing technology, researchers have been able to reconstruct the geometric models of coarse aggregates and achieve the quantitative characterization of morphological parameters [4]. For example, Aliha et al. quantified the angularity of aggregates through the fractal dimension and found that for every 10% increase in the angularity index, the compressive strength of the mixture increased by approximately 7% [5]. Wang et al. by combining the discrete element method-finite element method coupling model (DEM-FEM), revealed a strong correlation between the aspect ratio of aggregates and the crack propagation path [6]. E. Masad [7,8] used CT scanning technology to reconstruct the interior of asphalt concrete and found that the distribution of void sizes in the micro specimens of asphalt concrete is uneven, with a larger size at both ends and a smaller size in the middle. Li Fen [9] reconstructed the micro model that reflects the relationship between the various phases of the asphalt mixture through CT scanning technology and numerically simulated the crack development process of the specimen.

Although image processing technology can obtain a more realistic micro model, it requires a real specimen as a basis, and the process is time-consuming, labor-intensive, and costly, so its application is also limited. Random digital generation technology generates aggregates, voids, etc., through algorithms, controlling parameters such as aggregate shape, size, content, gradation, aggregate spacing, and void size. Wang et al. [10] proposed a random aggregate generation method based on the Monte Carlo random sampling principle at the end of the 20th century. Sun [11] proposed a concave-convex aggregate generation method and used the "double background grid method" to accurately and efficiently determine whether there is an overlap between aggregates, but it did not consider the generation of needle-shaped aggregates. Masad et al. [12,13] quantified the geometric characteristics of aggregates from the perspective of aggregate angle and aggregate surface texture through digital image processing technology. Xie et al. [14] analyzed the geometric characteristics of aggregates through image processing technology and quantified them through indicators, exploring the relationship between mixture performance and these geometric indicators. Jin et al. [15] used mathematical statistical ideas to reconstruct the three-dimensional entity of aggregates, classified them according to the angularity and surface characteristics of aggregates, and then

explored the relationship between the performance of asphalt mixtures and aggregate characteristics through numerical simulation. Wang Hai Nian et al. [16] studied the angularity of aggregates as the main geometric characteristic of aggregates and found that evaluating the angular characteristics of aggregates with fractal dimension and roughness can better reflect the morphological characteristics of aggregates. Jin Can et al. [17] reconstructed the three-dimensional model of aggregates using CT scanning technology and measured the sphericity, angularity, and volume of aggregates, and finally proved the rationality of this evaluation standard through experimental methods. Most scholars' research on the morphological characteristics of aggregates is based on image – processing technology [18–20]. After collecting the morphological characteristics of aggregates, they obtain the sensitivity of their characteristic parameters through statistical analysis. This technology is not only complicated to operate but also unable to conduct a comparative analysis of a single index. Moreover, few scholars have studied the morphology of recycled coarse aggregates in asphalt concrete.

In this paper, four parameters, namely the aspect ratio, shape parameter, roundness ratio, and angularity, are selected to quantitatively describe the external shape characteristics of aggregates. Based on the expressions of each characteristic parameter, criteria for randomly placing aggregates are established. Subsequently, two – dimensional finite element specimens with coarse aggregate parameters within a certain range are constructed. On the basis of referring to the research results of other scholars on the parameters of the cohesive zone model, indoor Marshall tests are carried out to verify the accuracy of the two – dimensional finite element meso-scale asphalt concrete finite element model. Through this model, the influence of different coarse aggregate morphologies on the mechanical properties of asphalt mixtures is deeply explored, and then the relationship between the meso-scale morphology of aggregates and the macroscopic mechanical properties is revealed.

## 2. Two-dimensional microscopic model

### 2.1. Selection of coarse aggregate morphological parameters

The aspect ratio, proposed by Kuo and Freeman [21] as an evaluation index for assessing aggregates in asphalt concrete using image analysis technology, has been used to explore the relationship between aggregate morphological characteristics and the performance of asphalt mixtures. They introduced several aggregate morphological feature parameters for evaluating the performance of aggregates. It can be represented by the Formula 1.

$$AR = \frac{L}{W}$$

(1)

In the formula, L represents the length of the aggregate image; W represents the width of the aggregate image.

The shape parameter, introduced by Masad [22], is derived from the image analysis and morphological feature assessment of fine aggregates in asphalt mixtures, as well as the study of the correlation between fine aggregate morphological parameters and the performance of asphalt mixtures. It is concluded that fine aggregate morphological parameters are one of the important factors in predicting the performance of asphalt mixtures, and this parameter can be used to assess the durability, reliability, and performance of asphalt mixtures.

Mora [23] proposed a method using digital image processing technology to measure the sphericity, shape factor, and convexity of coarse aggregates in concrete. The author put forward a measurement method based on digital image processing technology, which can rapidly and accurately measure and analyze the morphological characteristics such as sphericity, shape factor, and convexity of coarse aggregates in concrete.

Kuo [24] introduced a digital image processing index method for quantifying the shape, angularity, and surface texture of aggregates. Through processing digital images, a series of morphological characteristic indices, including angularity, shape, and surface texture, were proposed for the quantitative analysis of the morphological characteristics of aggregates.

This paper selects four mesoscopic characteristics, namely the aspect ratio, shape parameter, roundness ratio, and angularity, as the morphological characteristics of coarse aggregates for research. The morphological characteristics are characterized through the analysis of the geometric shapes of the aggregates.

## 2.2. Method for extracting coarse aggregate morphological parameters

The aggregates used in the test were derived from Reclaimed Asphalt Pavement (RAP) materials. According to Method T0722 - 1993 in JTG E20 - 2011, the aggregates obtained in the container after the asphalt solution was extracted by a centrifugal extractor were first evaporated in a fume hood or indoor air, and then dried in an oven at a temperature of 105°C±5°C.

To obtain aggregate images, aggregates are categorized and collected based on particle size ranges such as 2.36 mm to 4.75 mm, 4.75 mm to 9.5 mm, 9.5 mm to 13.2 mm, and 13.2 mm to 16 mm for image acquisition. Subsequently, Photoshop is utilized to enhance the image effects and adjust the contrast, ensuring the clarity and visibility of the aggregate features. The required parameters for measurement are then selected, and the Image-Pro Plus (IPP) software is employed for analysis to derive the necessary parameter data. This process aims to enhance the precision of subsequent software in recognizing and analyzing the images. As shown in Fig 1.

## 2.3. Determination of aggregate placement and gradation

Based on the AC-16 gradation as the aggregate placement gradation, the area is used as the controlling factor. Taking the specimen area SS as an example, the required placement area for each size of aggregate is calculated through the following formula, where pp represents the percentage by number passing, and dd is the sieve size.

$$S_i = S \cdot \frac{(p_i - p_{i+1}) \cdot \left(\frac{d_{i+1}+d_i}{4}\right)^2 \pi}{\sum_{j=1}^{5} (p_j - p_{j+1}) \cdot \left(\frac{d_{j+1}+d_j}{4}\right)^2 \pi}$$

(2)

Taking the two-dimensional Marshall specimen as an example, the required material placement areas for each gradation level in the Marshall Splitting Test specimen can be determined as shown in Table 1.

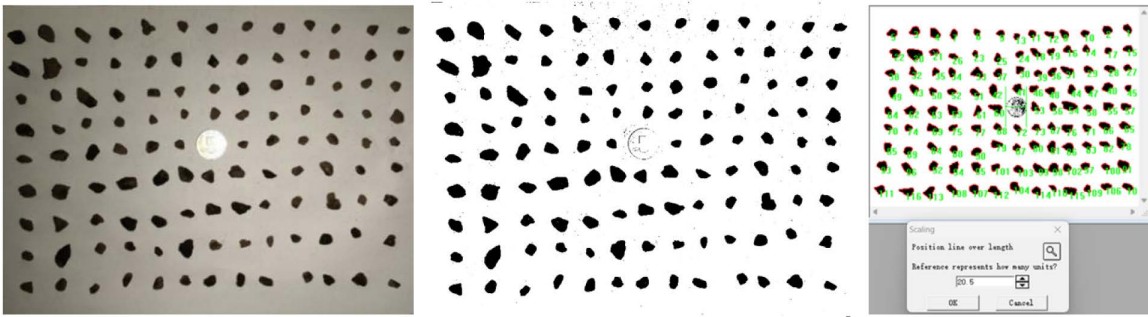

**Fig 1. Aggregate Measurement Image in IPP.**

**Table 1. Splitting Specimen Aggregate Placement Area.**

| Particle Size Range/mm | 2.36~4.75 | 4.75~9.5 | 9.5~13.2 | 13.2~16 | 16~19 |
|---|---|---|---|---|---|
| Placement Area/mm² | 1164 | 1336 | 847 | 841 | 674 |

## 2.4. Random generation and placement of aggregates

In this study, the Python language built into ABAQUS software is utilized to compile code for the automatic placement of aggregates, with the goal of generating randomly shaped aggregates that are evenly distributed. Through multiple trials of random placement, it has been found that the number of edges for aggregates in the asphalt concrete microscopic model should be between 8–15 edges to be optimal. This range provides the necessary plasticity for the aggregates while preventing the generation speed from becoming too slow due to an excessive number of edges. The specific placement process is illustrated in the following Fig 2.

(1) A center of the circle is randomly generated according to the specimen radius R and the maximum diameter "$d_i$" of each gradation, and it serves as the starting point for generating aggregates.

$$x = \text{random. uniform}(-R,R) \tag{3}$$

$$y = \text{random. uniform}(-R,R) \tag{4}$$

Meanwhile, restrictions are imposed on the generated values of *x* and *y*.

$$x^2 + y^2 \le (R - \frac{d_i}{2})^2 \tag{5}$$

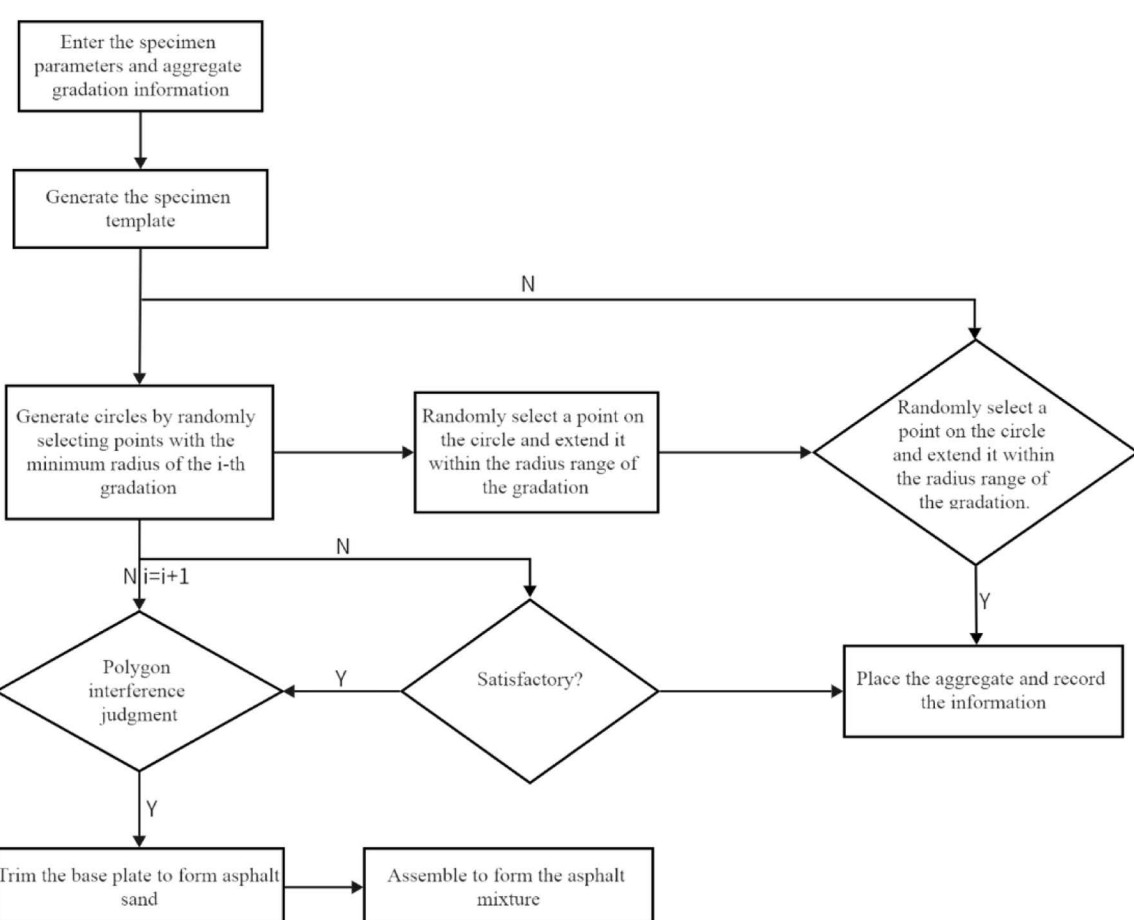

**Fig 2. Flowchart of Random Aggregate Placement Process.**

To ensure the plasticity of aggregates, a circle is generated with the center at the point (x, y) obtained above and the minimum diameter $d_{i+1}$ of each gradation as the diameter, providing an initial range for generating the vertices of polygonal aggregates. The number of sides of the polygon is randomly generated to be in the range of 8–15, specifically, e = random. randint(8, 15). In order to obtain aggregates with uniform shapes, the circle is divided into 3 × e parts here. The selection range of adjacent vertices is spaced by one part of the area. Taking the angle θ between the ray from the center of the circle to the vertex of the aggregate and the positive direction of the x – axis as a parameter, an angle is randomly generated within the specified area, as shown in Fig 3. Then the coordinates of the vertices are obtained.

$$X = x + d_{i+1} \times \cos\theta \tag{6}$$

$$Y = y + d_{i+1} \times \sin\theta \tag{7}$$

(3) Extend the vertex coordinates generated in the previous step. The default extension range is "0~ $(d_i\text{-}d_{i+1})$/2". During the aggregate extension process, it is found that after the large – sized aggregates are generated, due to the significant difference between their diameters and the extension range, the plasticity during the generation of large aggregates is extremely poor. Most of the aggregates are in a nearly circular shape with uniform morphology, which cannot meet the parameters for controlling aggregates in the following text. Therefore, in order to meet the requirements for shaping aggregates in the following text, the extension range of aggregates with a particle size larger than 9.5 mm needs to be expanded, as shown in Fig 4. The expanded range is: $(d_{i+1}\text{-}d_i)$/2~ $(d_i\text{-}d_{i+1})$/2. For aggregates with a particle size of less than 9.5 mm, since the ratio of the aggregate particle size to the extension range is relatively large, there is no need to increase the extension range. Here, the extension range is set as follows: 0~ $(d_i\text{-}d_{i+1})$/2.

(4) Based on the vector operations in GIS (Geographic Information System) algorithms, compile with Python to implement interference judgment for aggregates and calculation of aggregate areas.

## 2.5. Characteristic parameters and geometric construction of coarse aggregates

(1) Geometric construction of the aspect ratio

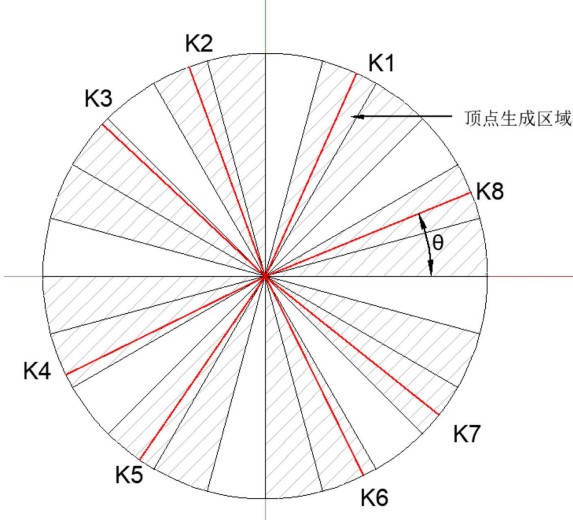

**Fig 3. Schematic diagram of the generation of polygon vertices.**

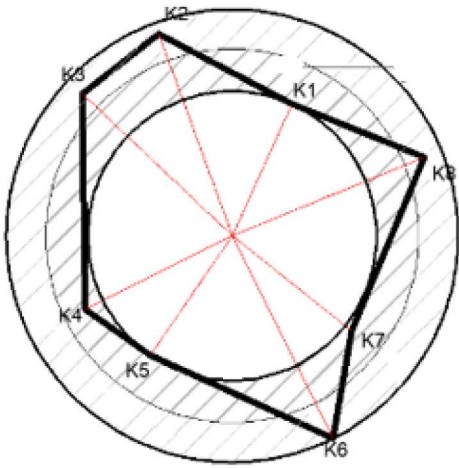 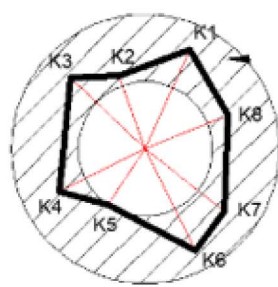

**Fig 4. Schematic diagram of the extension of vertices of large and small polygons.**

To control the aspect ratio of polygonal aggregates, on the basis of the aggregate generation described above, it is necessary to add restrictive conditions after the vertices of the polygon are generated. The Shapely library in Python is used to calculate the minimum bounding rectangle of the polygon. The vertex coordinates of the input polygon are converted into a Polygon object in Shapely, and then the convex_hull method is called to obtain the convex hull of the polygon, which is the smallest convex polygon containing the given polygon. Subsequently, the minimum_rotated_rectangle method is called to obtain the minimum rotated rectangle of the convex hull, that is, the smallest rectangle that can enclose the convex hull [25]. Finally, the four corner points of this rectangle are obtained, and thus the minimum bounding rectangle of the polygon is obtained, and the aspect ratio of this rectangle can be calculated.

Fig 5a shows the aggregates generated under the restrictive conditions of aspect ratios of 1.75 to 1.7, 1.6 to 1.55, 1.45 to 1.4, 1.3 to 1.25, and 1.15 to 1.1 respectively. It can be observed that as the aspect ratio gradually decreases, the shape of the aggregates changes from a "slender shape" to a "square shape."

(2) Geometric construction of the shape parameter

To restrict the shape parameter of polygonal aggregates, the restrictive conditions are changed, and the restriction function of the shape parameter is used for constraint. In this paper, the cross to product method is used to calculate the area of the polygon [26]. When the restriction range of the shape parameter is relatively small, the generated aggregate shapes have obvious edges and irregularities. As the parameter restriction range gradually increases, the shapes of the

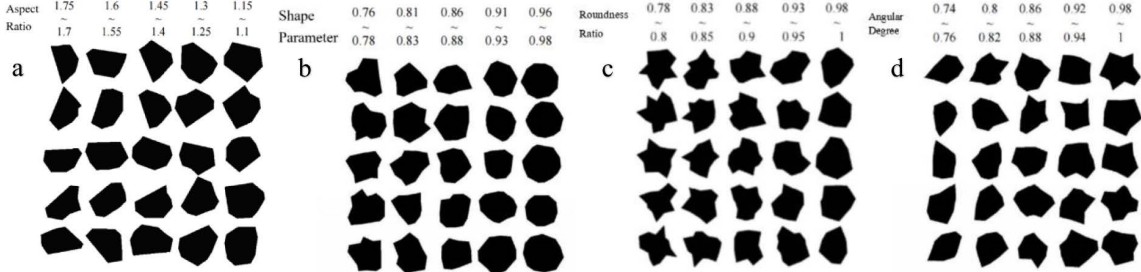

**Fig 5. Example Diagram.**

aggregates become more rounded and regular. The Fig 5b shows the aggregates generated under the shape parameter restrictions of 0.76 to 0.78, 0.81 to 0.83, 0.86 to 0.88, 0.91 to 0.93, and 0.96 to 0.98 respectively.

(3) Geometric construction of the roundness ratio

In this paper, the roundness ratio, a two to dimensional aggregate evaluation index, is selected as the research object [27]. In particular, when the aggregate image is a circle, an ellipse, or an equilateral polygon, its roundness ratio is 1. The calculation of the convex boundary area is based on Andrew's monotone chain algorithm, which is an efficient algorithm for calculating the convex hull. First, the point set is sorted according to the coordinates. Then, the point set is traversed, and each point is pushed onto the stack in turn. If the pushed point destroys the convexity of the stack, the top elements of the stack will be continuously popped until the stack becomes convex again. After traversing the point set, the elements in the stack are the vertices of the convex hull of the polygon, and the area of the convex hull can be calculated using these vertices. The Fig 5c shows the placed aggregates when the roundness ratio is controlled within the ranges of 0.78 to 0.8, 0.83 to 0.85, 0.88 to 0.9, 0.93 to 0.95, and 0.98 to 1 respectively.

(4) Geometric construction of angularity

The least to squares method is used to fit an ellipse, and the perimeter of the equivalent ellipse is calculated based on the major and minor axes of the fitted ellipse [28]. First, the centroid of the polygon is found. Then, the point farthest from the centroid is located, and the distance between this point and the centroid is defined as the major axis a of the ellipse. Subsequently, the minor axis b of the ellipse is obtained by dividing the area of the polygon by the value of π. The Fig 5d shows the aggregates within different control ranges of the angularity index, which are 0.74 to 0.76, 0.8 to 0.82, 0.86 to 0.88, 0.92 to 0.94, and 0.98 to 1 respectively.

## 3. Establishment of 2D digital specimens and acquisition of material parameters

### 3.1. Establishment of 2D digital specimens

Based on the random aggregate placement algorithm and the specimens from indoor tests, a finite element model is established. As shown in Fig 6. A circular specimen with a radius of 50.8 mm and a rectangular specimen with a length of 101.6 mm and a width of 63.5 mm are respectively established according to the compressive test and the splitting test.

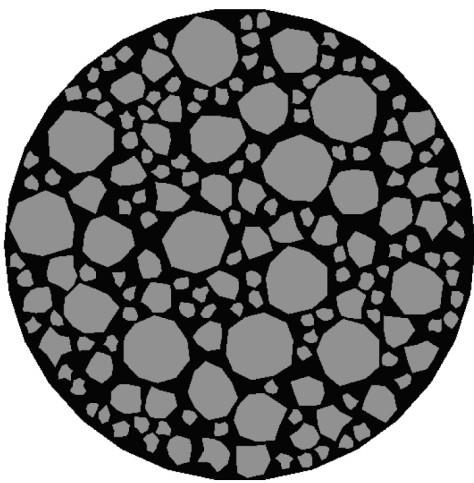
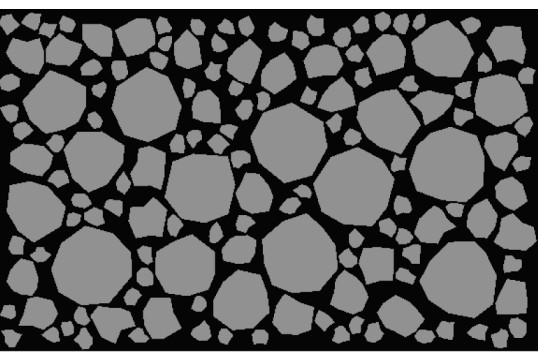

**Fig 6. Specimen for the splitting test.**

By indexing the elements for targeted selection, it is only necessary to set a parameter during the aggregate placement process to calculate the number of aggregates larger than 4.75 mm in diameter to quickly select the required objects. The aggregates are split by edges, and the planting is controlled by the number of elements, with one element per edge. Elsewhere, global planting is used with an approximate size of 0.5 for planting. After meshing, the total number of elements is 49,896, and the meshing results are shown in Fig 7.

### 3.2. Acquisition of material parameters

**3.2.1. Boundary conditions of the compressive test.** In order to be consistent with the indoor compressive test, in this section, the loaders of the simulated specimens are set as a rectangular loader with dimensions of 120 mm × 10 mm at the upper and lower parts respectively, as shown in Fig 8. In Abaqus, the upper and lower loaders are set as rigid bodies. The lower loader is fixed, and a downward displacement is applied to the upper loader. The loading amplitude is set as a linear function so that the displacement can be loaded at a constant speed.

The compressive test is carried out using a compression testing machine, and the loading rate is controlled at 2 mm/min.

According to the "Test Regulations for Asphalt and Asphalt Mixtures in Highway Engineering" of our country, the compressive strength can be calculated by the following formula:

$$R_c = \frac{4P}{\pi d^2}$$

(8)

$R_c$ represents the compressive strength of the Marshall specimen (MPa).
$P$ represents the maximum load (N) when the specimen is damaged.
$d$ represents the diameter of the specimen (mm).

**3.2.2. Boundary conditions of the splitting test.** To correspond with the indoor splitting test, the loading device for the simulated specimen in this section is set with one above and one below, each being a rectangular loader with

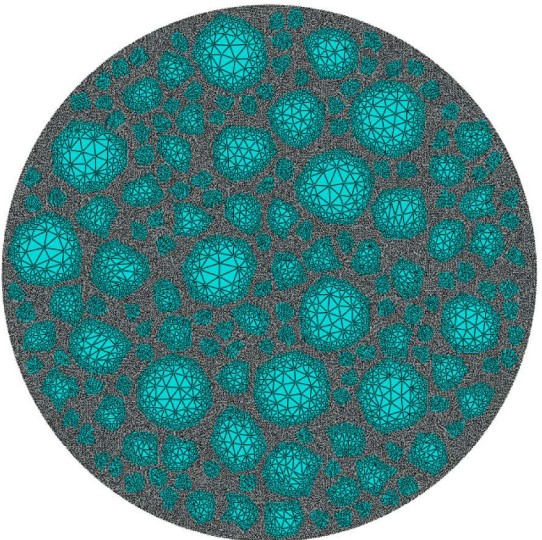 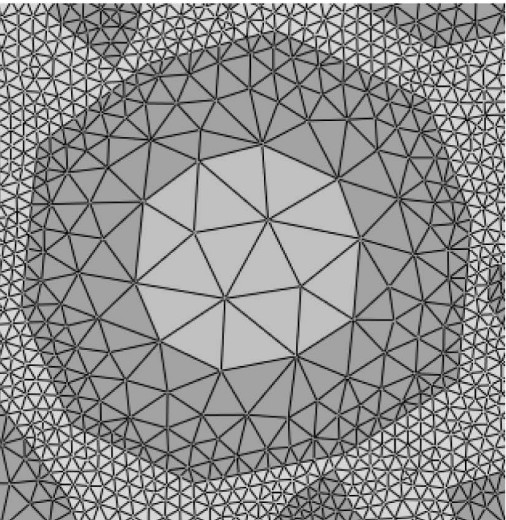

**Fig 7. Mesh Division Schematic Diagram.**

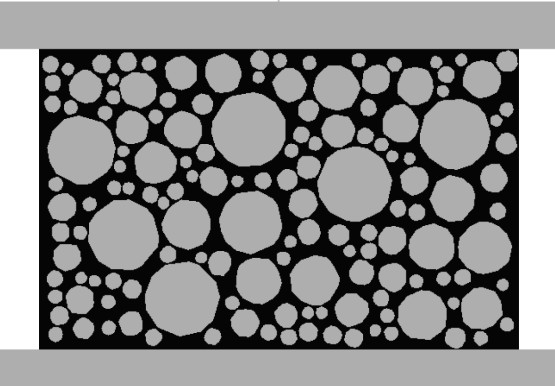

**Fig 8. Distribution of loaders in the compressive test.**

dimensions of 12.7 mm by 10 mm. In Abaqus, the upper and lower loaders are set as rigid bodie. The loading amplitude is set as a linear function to ensure that the displacement is applied at a constant velocity. As shown in Fig 9.

The Marshall Splitting Test is conducted using a pressure testing machine. The Marshall specimens are placed in a constant temperature water bath at 15°C for 3 hours to equilibrate. After removal, they are quickly placed into the splitting test fixture, followed by applying a loading rate of 50 mm/min.

According to the "Highway Engineering Asphalt and Asphalt Mixture Test Specifications" the tensile strength can be calculated using the following formula 9.

$$R_T = \frac{0.006287P_T}{h}$$

(9)

In the formula, $R_T$ stands for the splitting tensile strength (MPa); $P_T$ is the maximum load at the time of specimen splitting failure (N); $h$ represents the height of the Marshall specimen (mm).

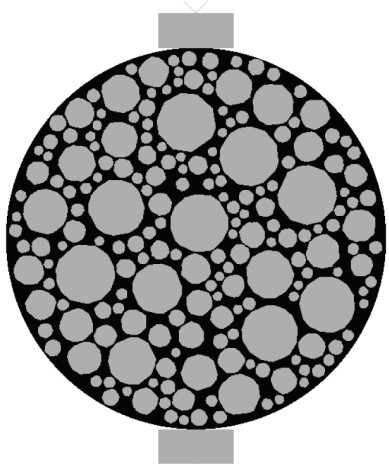

**Fig 9. Distribution of Splitting Test Loaders.**

This paper refers to the research results of Kollmanna [29] to fit the aforementioned test parameters and ultimately selects the micro parameters as shown in Table 2.

Using the aforementioned material parameters, the maximum load at model failure is calculated as shown in Table 3.

The simulated specimens used for fitting with indoor tests did not impose very strict characteristic parameter restrictions on the aggregates. As a result, the data from the simulated splitting test is lower than the measured values from the actual indoor tests, with a difference of 2.2 KN. The value measured in the compressive simulation experiment is higher than that measured in the actual indoor experiment, with a difference of 2.67 KN. Looking at the comparison between the test results and the simulation results, these parameters can reflect the comprehensive failure capacity of the asphalt mixture quite well.

## 4. Results and discussion

Based on the random aggregate placement algorithm and the relevant parameters of the meso-scale model of asphalt concrete mixture, the mechanical property models of asphalt mixture under different characteristic parameters of coarse aggregates are established, and the influence of each characteristic parameter on the mechanical properties of asphalt mixture is analyzed. The placement area of aggregates accounts for 60% of the total area, and the gradation adopts the placement proportion of the two-dimensional AC-16 specimens converted in the previous text.

The approximate distribution of the internal stress nephogram of the specimen during loading is shown in Fig 10.

### 4.1. Analysis of the influence of coarse aggregate morphology on the compressive performance of asphalt mixture

**4.1.1. Analysis of the influence of aspect ratio on compressive performance.** Taking the compression process of the specimen with an aspect ratio of 1.55 to 1.60 as an example, observe the internal damage changes of the specimen, As shown in Fig 11.

During the loading process of the specimen with an aspect ratio of 1.55 to 1.60, the initial cracks are generated uniformly and are relatively small. The cracks almost occur at the two ends of the aggregate parallel to the horizontal axis. Judging from the distribution of the cracks, the compactness of the aggregates inside this specimen is relatively good, and the forces at various parts of the specimen are relatively uniform. Cracks appear simultaneously at multiple locations inside the specimen. At this time, the displacement of the specimen is relatively large, so the rate of energy change is relatively small. As shown in Fig 12.

After the initial cracks are generated, obvious cracks appear at multiple locations inside the specimen. The cracks are relatively short in length and evenly distributed, indicating that the internal forces of the specimen are relatively uniform,

**Table 2. Material Parameters.**

| Material | Elastic Modulus (Mpa) | Poisson's Ratio | Fracture Energy (Nmm/mm²) | Bond Strength (Mpa) |
|---|---|---|---|---|
| Coarse Aggregate | 20000 | 0.15 | — | — |
| Asphalt Mortar | 5240 | 0.3 | — | — |
| Mortar-Mortar Interface | 5240 | — | 1.631 | 2.5 |
| Mortar-Coarse Aggregate Interface | 5240 | — | 1.631 | 2.3 |

**Table 3. Ultimate Failure Load.**

| Name | Splitting Test | Simulated Splitting Test | Compressive Test | Simulated Compressive Test |
|---|---|---|---|---|
| Average Ultimate Failure Load (KN) | 13.56 | 11.36 | 35.24 | 37.92 |

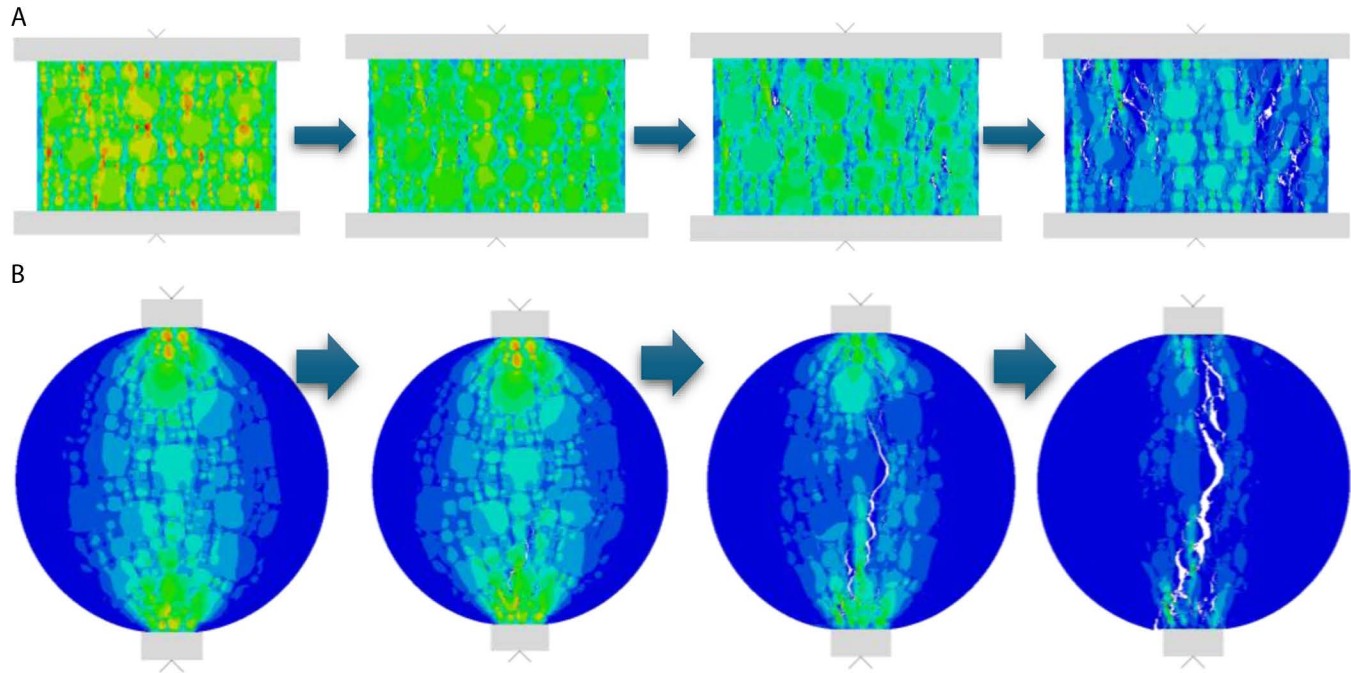

**Fig 10. Stress Nephogram of the Splitting Specimen Process.**

**Fig 11. Internal changes of the specimen 1.55 to 1.60.**

and there is no problem of excessive local stress. Therefore, the internal cracks do not extend too long in a certain place. At this time, the displacement of the specimen is relatively large, so the rate of energy change is relatively fast.

When the specimen breaks, multiple cracks appear inside, reaching the highest bearing capacity of the specimen. At this time, the displacement of the specimen is relatively large, so the rate of energy change is relatively fast, and the total energy consumption of the specimen during failure is relatively large.

As the aspect ratio decreases, the energy consumed when the specimen fails exhibits a trend of rising first and then falling. When the aspect ratio is between 1.40 and 1.45, the energy consumed during specimen failure reaches its maximum value of 44.88 J; when the aspect ratio is between 1.0 and 1.15, the energy consumption for specimen failure attains its minimum value of 36.63 J.

Concerning the compressive strength, as the aspect ratio rises, it follows a pattern of increasing initially and then decreasing. Among all the specimen groups, when the aspect ratio is between 1.55 and 1.60, the compressive strength peaks at 5.06 MPa; when the aspect ratio is between 1 and 1.15, the compressive strength bottoms out at 4.48 MPa.

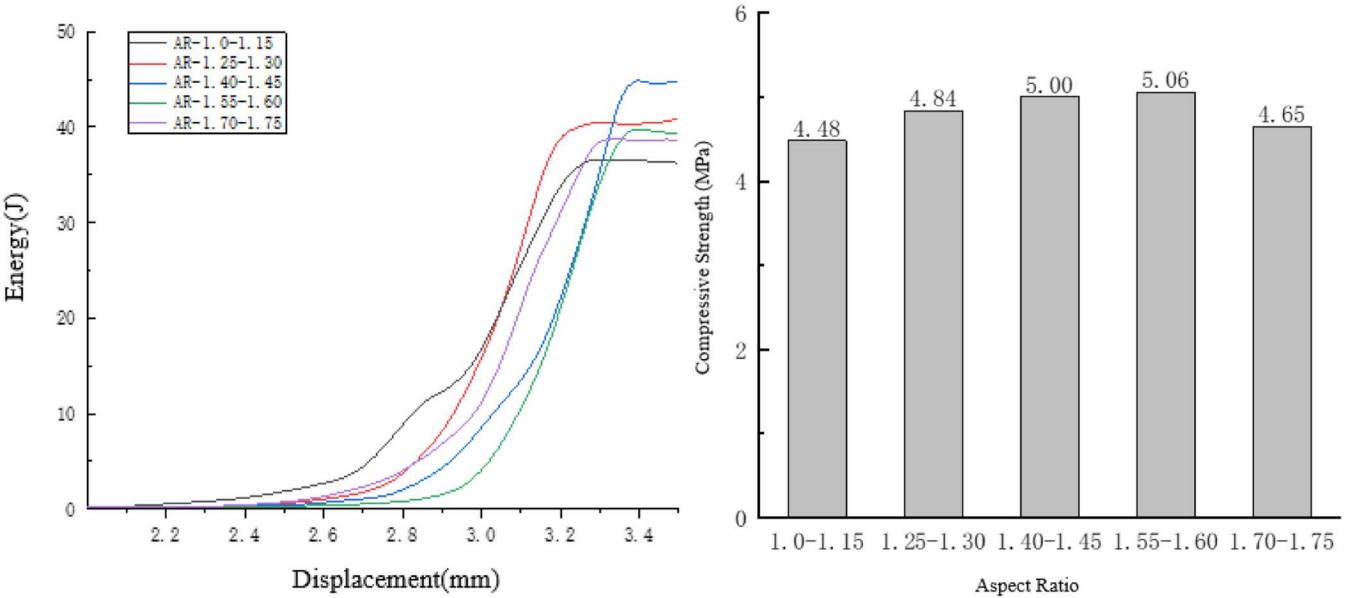

**Fig 12. Influence of the Aspect Ratio on the Compressive Performance of the Specimens.**

**4.1.2. Analysis of the influence of shape parameters on compressive performance.** Taking the compression process of the specimens with shape parameters ranging from 0.70to 0.75 as an example, observe the internal damage changes of the specimens. As shown in Fig 13.

For the specimens with shape parameters ranging from 0.7 to 0.75, initial cracks only occurred locally. Moreover, these cracks were significantly larger than those in other parts and their distribution was more concentrated. Therefore, the rate of initial damage change was relatively large. The cracks of the specimens extended upward and downward at a 45-degree angle, and only one large crack appeared inside the specimen while small cracks emerged in other areas. This was due to the fact that the aggregate outline was too irregular, resulting in larger defects and more concentrated stress at that location.

Thus, the rate of change during the damage development stage was relatively large. Eventually, the cracks of the specimen extended at a 45-degree angle in the horizontal direction, and a small number of short cracks appeared in the remaining parts of the specimen. As shown in Fig 14. At this time, the specimen had lost its bearing capacity, so the total energy consumption of the specimen was relatively low. It can be seen that when the shape parameter is too small, the bearing capacity of the specimen decreases due to uneven internal stress. However, overall, the influence of the shape parameter on its compressive performance is not prominent.

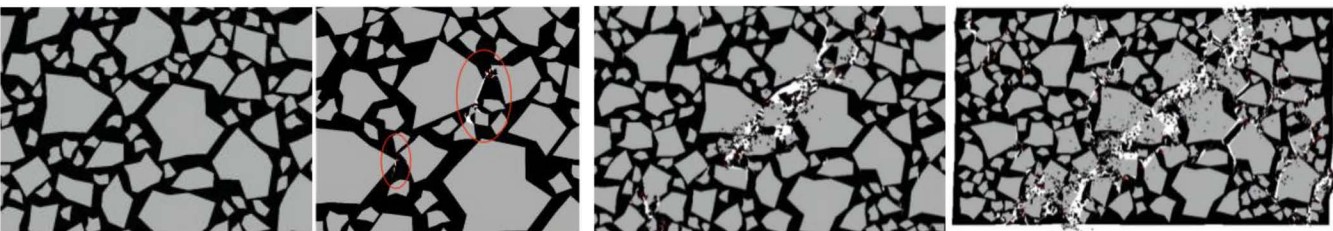

**Fig 13. Internal changes of the specimen of 0.7 to 0.75.**

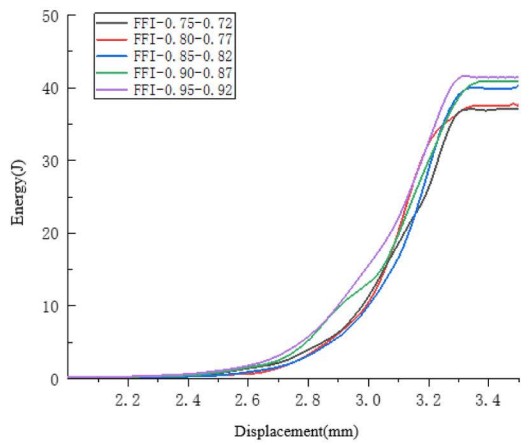
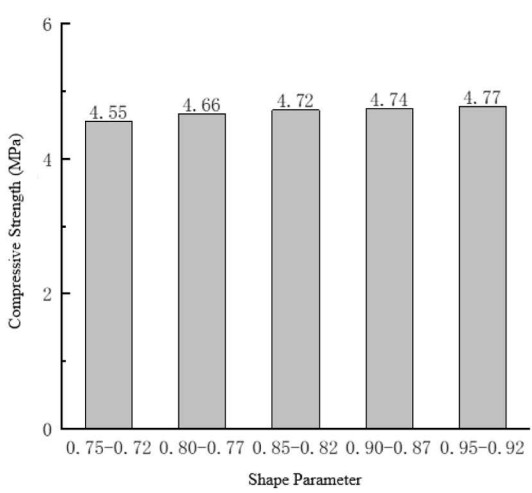

**Fig 14. Influence of Shape Parameters on the Compressive Performance of the Specimens.**

As the shape parameter decreases, the compressive strength of the specimen shows a downward trend. Among them, when the shape parameter is between 0.92 and 0.95, the specimen reaches the maximum compressive strength of 4.77 MPa, and when the shape parameter is between 0.75 and 0.72, the specimen reaches the minimum compressive strength of 4.55 MPa. As the shape parameter decreases, the energy consumed when the specimen fails also shows a downward trend.

**4.1.3. Analysis of the influence of roundness ratio on compressive performance.** Taking the compression process of the specimens with a roundness ratio ranging from 0.98 to 1 as an example, observe the internal damage changes of the specimens. As shown in Fig 15.

For the specimens with a roundness ratio ranging from 0.89 to 1, the initial cracks mainly appear at the protruding positions at the horizontal ends of the aggregates. At this time, the displacement of the specimen is small, so the rate of energy change is small.

Due to the uneven distribution of its initial cracks and the large disparity in crack widths, the cracks mainly develop along the initial cracks. There is significant local damage to the specimen, while its displacement is small. The overall bearing capacity of the specimen decreases at a relatively slow pace, resulting in a relatively small rate of energy change. As shown in Fig 16. In the end, the crack distribution of the specimen is mainly concentrated at both ends, showing a relatively concentrated pattern. The local damage to the specimen is rather severe, and consequently, its rate of energy change is slow.

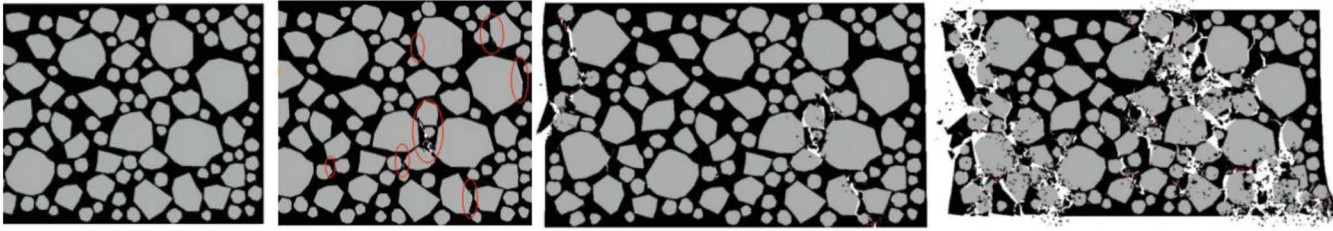

**Fig 15. Internal changes of the specimen of 0.89 to 1.**

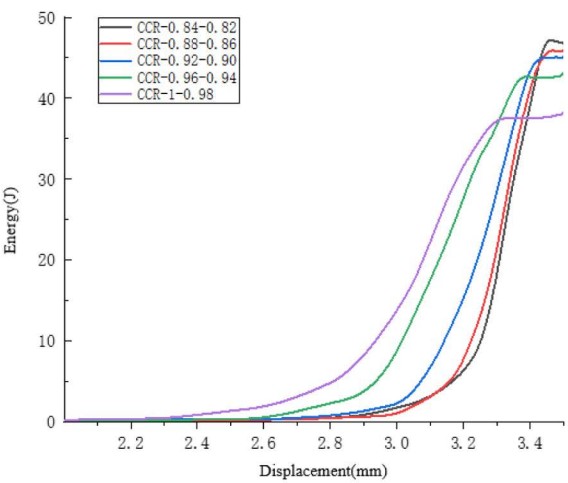
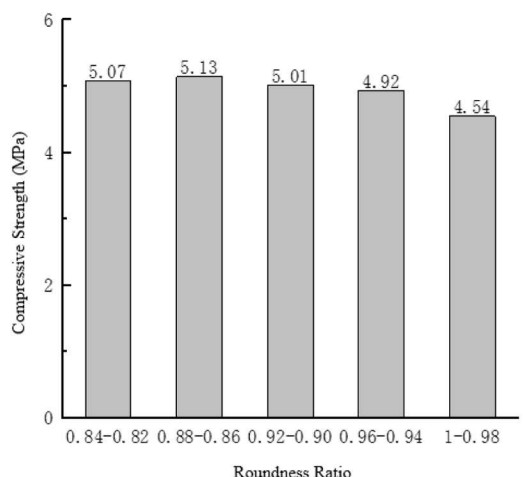

**Fig 16. Influence of the Roundness Ratio on the Compressive Performance of the Specimens.**

As the roundness ratio increases, the energy consumed when the specimen fails presents a downward trend. When the roundness ratio is in the range of 0.98 to 1, the energy consumed for the specimen failure reaches the minimum value of 38.35 J.

With the increase in the roundness ratio, the compressive strength of the specimen generally shows a downward trend. When the roundness ratio is between 0.86 and 0.88, the specimen achieves the maximum compressive strength of 5.13 MPa. When the roundness ratio is between 0.98 and 1, the specimen reaches the minimum compressive strength of 4.54 MPa.

**4.1.4. Analysis of the influence of edge angle on compressive performance.** Taking the compression process of the specimens with edge angle ranging from 0.86 to 0.88 as an example, observe the internal damage changes of the specimens. As shown in Fig 17.

In the overall specimens, for the specimens with an edge angle ranging from 0.86 to 0.88, the initial cracks are generated at both ends and the middle of the specimens. This indicates that an excessively small edge angle will lead to uneven internal forces in the specimens, affecting their bearing capacity.

After the initial cracks appear, the crack distribution is relatively uniform. Long cracks with little difference in thickness appear at both ends and in the middle of the specimens, and the specimens suffer from overall damage. Therefore, the trend of energy change is relatively fast.

As shown in Fig 18. The influence of the long-axis directivity of the aggregates with a low edge angle on the overall bearing capacity of the specimens: When relatively uniform cracks appear inside the specimens, and cracks are present

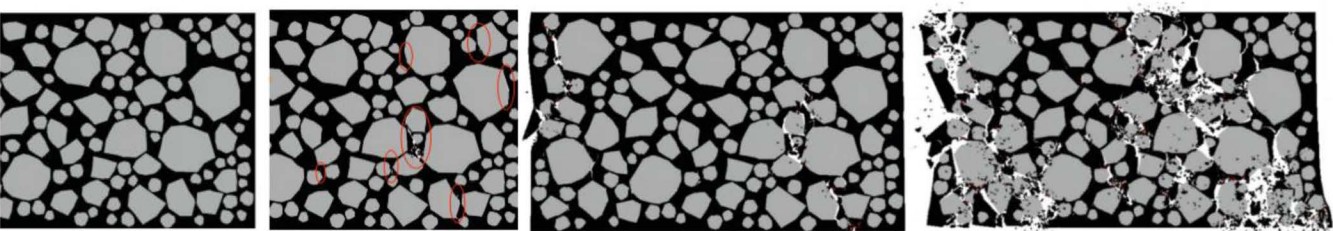

**Fig 17. Internal changes of the specimen of 0.86 to 0.88.**

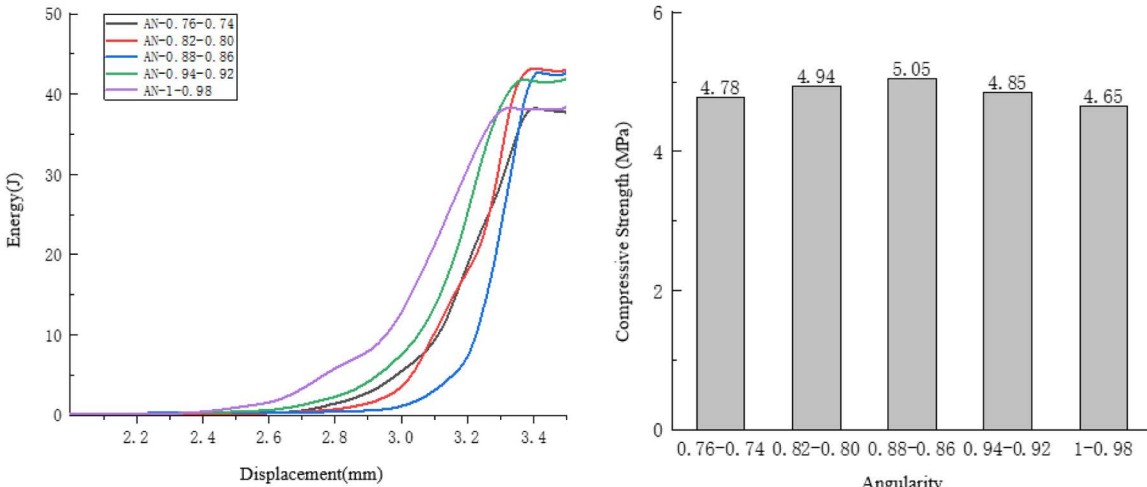

**Fig 18. Influence of Angularity on the Compressive Performance of the Specimens.**

at both ends and in the middle of the specimens, the specimens show overall damage, so the rate of energy change is relatively fast.

As the edge angle decreases, the compressive strength of the specimen shows a trend of first increasing and then decreasing. Along with the decrease in the edge angle, the energy consumed when the specimen fails also exhibits a trend of first rising and then falling. Among them, when the edge angle ranges from 0.80 to 0.94, the energy dissipation gradient is relatively smooth, and the energy consumption difference of the specimens is within 1 J.

### 4.2. Analysis of the influence of coarse aggregate morphology on the anti-splitting performance of asphalt mixture

**4.2.1. Analysis of the influence of aspect ratio on anti-splitting performance.** Taking the compression process of the specimens with an aspect ratio ranging from 1.70 to 1.75 as an example, observe the internal damage variation process of the specimens in Fig 19.

In the initial stage, fine cracks appear both at the upper and lower parts of the specimen. Moreover, the initial damage occurs at the position where the "convex edges face inward" of the aggregates along the loading direction, and the rate of energy change is relatively fast.

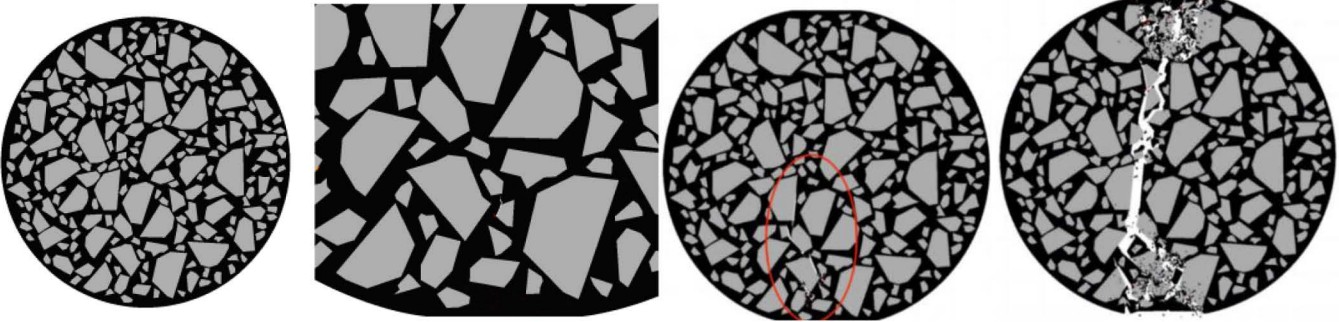

**Fig 19. Internal changes of the specimen of 1.70 to 1.75.**

After the initial cracks are generated, as the long edges of the aggregates develop along the loading direction, the displacement of the specimen is relatively large at this time, and the rate of energy change is also relatively fast.

The final failure cracks of the specimen deviate from the loading path and cause damage in the upward direction, which further proves that the aspect ratio can affect the crack resistance strength of the specimen. At this moment, the displacement of the specimen is large, so the rate of energy change is relatively fast. As shown in Fig 20.

As the aspect ratio increases, the tensile strength of the specimen generally shows an upward trend. There is no obvious pattern in the variation range of the tensile strength of adjacent groups of specimens, and the maximum variation range is 1.1 MPa. With the increase of the aspect ratio, the energy consumed during the failure of the specimen also shows a trend of first increasing and then decreasing.

**4.2.2. Analysis of the influence of shape parameters on anti-splitting performance.** Taking the compression process of the specimens with shape parameters ranging from 0.87 to 0.9 as an example, observe the internal damage variation process of the specimens. As shown in Fig 21.

The aggregates of the specimens with shape parameters ranging from 0.87 to 0.9 are relatively regular. The initial cracks occur at the convex parts of the aggregates, and the local damage of the specimens is severe. Therefore, the rate of energy change is slow.

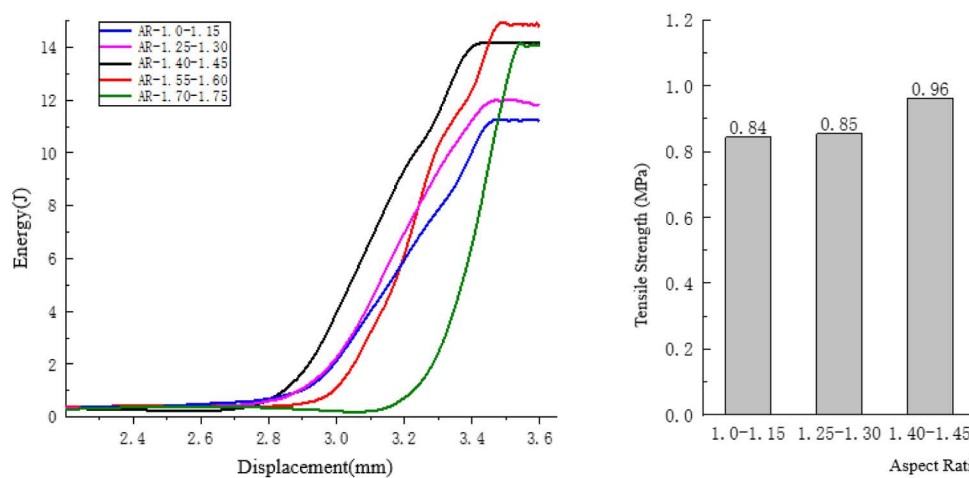

**Fig 20. Influence of the Aspect Ratio on the Splitting Resistance Performance of the Specimens.**

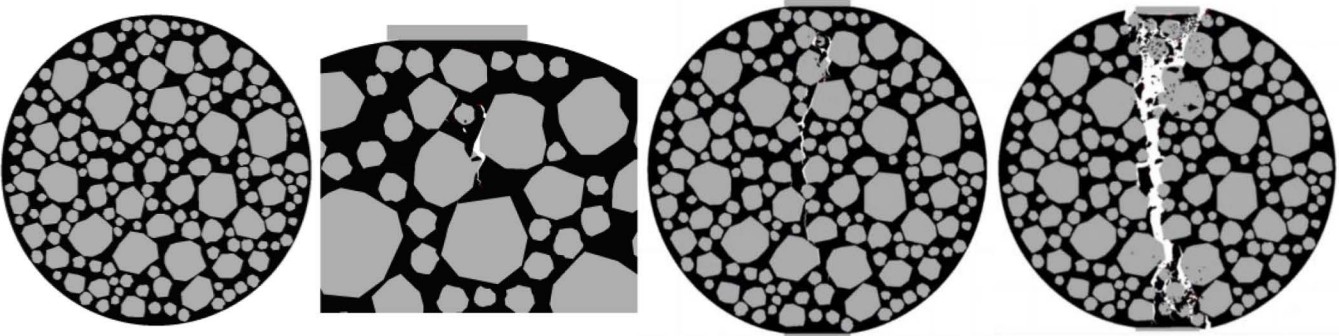

**Fig 21. Internal changes of the specimen of 0.87 to 0.9.**

After the initial cracks are generated, the cracks develop downward along the initial crack locations, and very few cracks are generated at other positions inside the specimens. At this time, the displacement of the specimens is small, so the rate of energy change is slow.

As shown in Fig 22. During the final failure, the cracks are relatively uniform. Due to the relatively smooth surface of the aggregates, the crack development is smooth, and thus the rate of energy change is slow.

As the shape parameter increases, the tensile strength of the specimen shows a trend of first increasing and then decreasing, while the energy consumed during the failure of the specimen generally exhibits a downward trend. The minimum energy consumption for specimen failure is 11.50 J.

### 4.2.3. Analysis of the influence of roundness ratio on anti-splitting performance.
Taking the compression process of the specimens with a roundness ratio ranging from 0.82 to 0.84 as an example, observe the internal damage changes of the specimens. As shown in Fig 23.

The initial cracks of the specimen are continuous, and the generation of the cracks is relatively smooth, so the rate of energy change is slow. The development path of the cracks extends along the loading direction. The crack development path is short and relatively smooth, so the rate of energy change is slow. Finally, the overall cracks are relatively straight and the crack path is short. However, at this time, the displacement of the specimen is large, so the rate of energy change is fast. As shown in Fig 24.

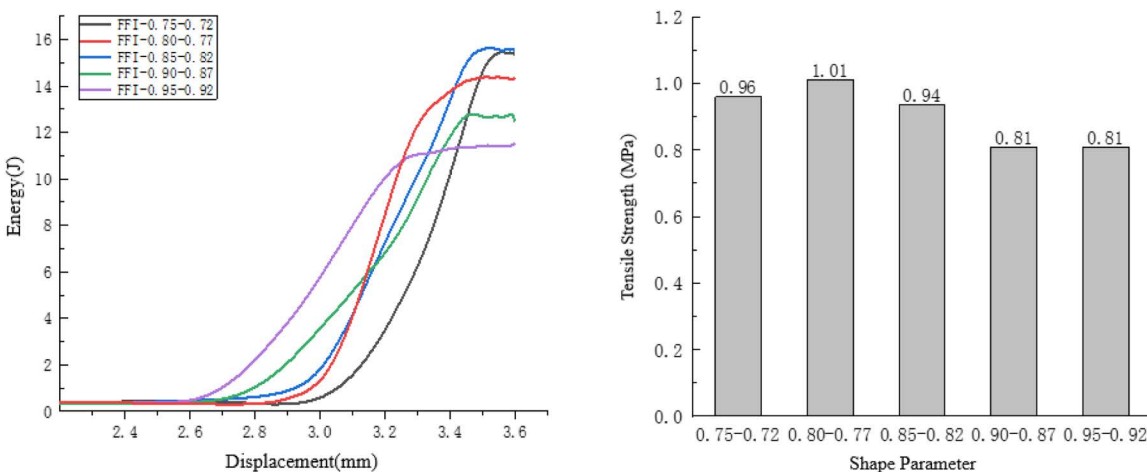

**Fig 22. Influence of Shape Parameters on the Splitting Resistance Performance of the Specimens.**

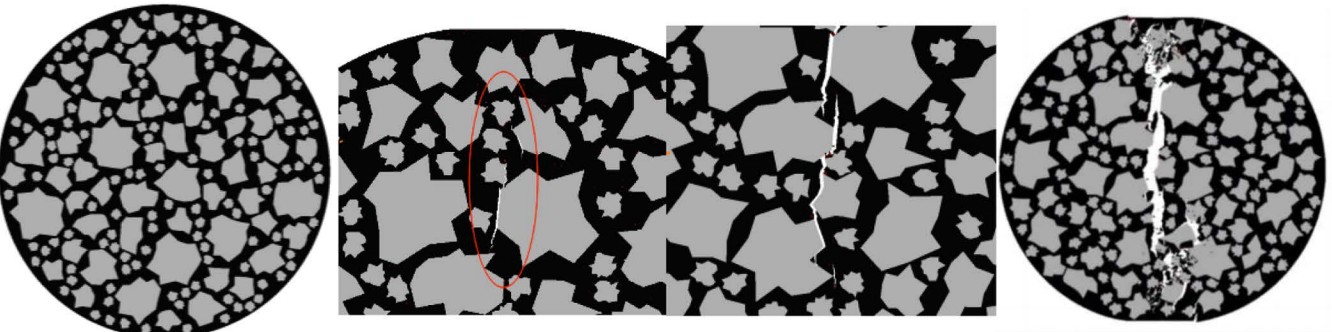

**Fig 23. Internal changes of the specimen of 0.82 to 0.84.**

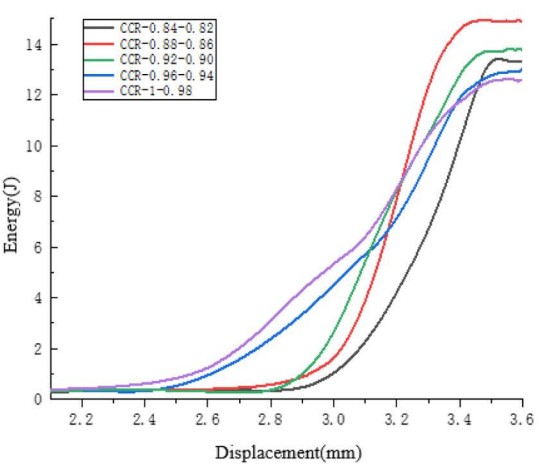 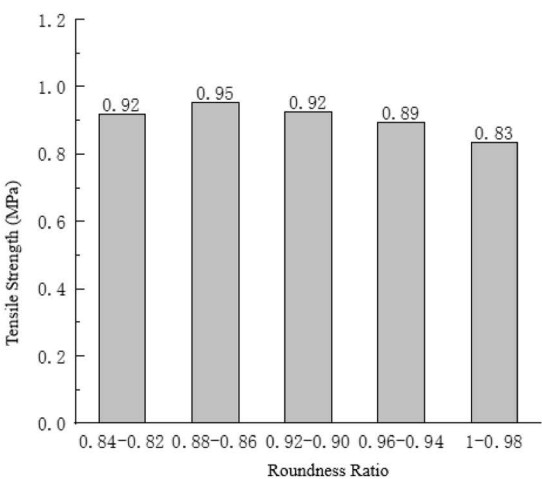

**Fig 24. Influence of the Roundness Ratio on the Splitting Resistance Performance of the Specimens.**

As the roundness ratio increases, the tensile strength of the specimen generally shows a trend of first increasing and then decreasing, and the energy consumed during the failure of the specimen also exhibits a trend of first rising and then falling.

**4.2.4. Analysis of the influence of angularity on anti-splitting performance.** Taking the compression process of the specimens with an angularity ranging from 0.74 to 0.76 as an example, observe the internal damage changes of the specimens. As shown in Fig 25.

Fine cracks appear simultaneously at multiple locations of the specimen, and all the cracks are located at the tip positions of the aggregates, so the rate of energy change is relatively fast.

Due to the initial cracks, the cracks generated in the specimen are relatively uniform, and multi-point fractures occur inside the specimen. The development of the cracks is relatively discontinuous, so the rate of energy change is relatively fast.

Cracks are generated at multiple locations both during the initial generation and the development of the cracks. At this time, the displacement of the specimen is large, so the rate of energy change is relatively high. As shown in Fig 26.

As the angularity increases, the tensile strength of the specimen shows a downward trend, and the overall energy consumed during the failure of the specimen also exhibits a downward trend.

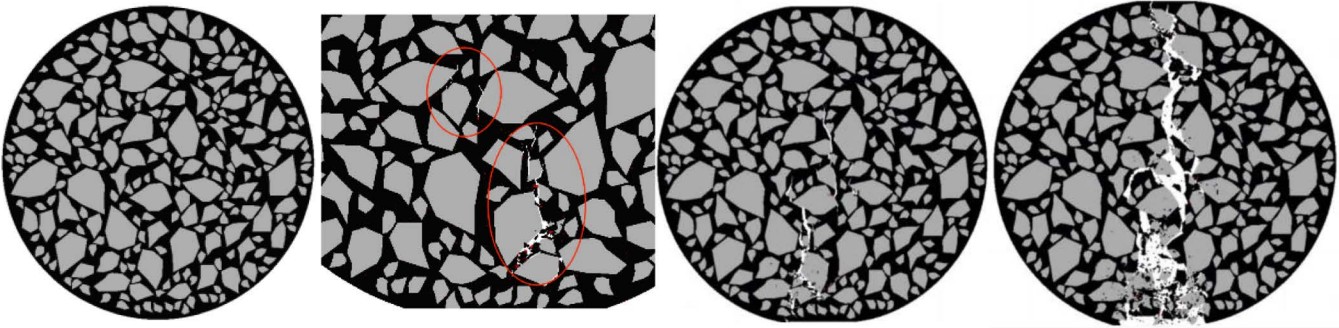

**Fig 25. Internal changes of the specimen of 0.74 to 0.76.**

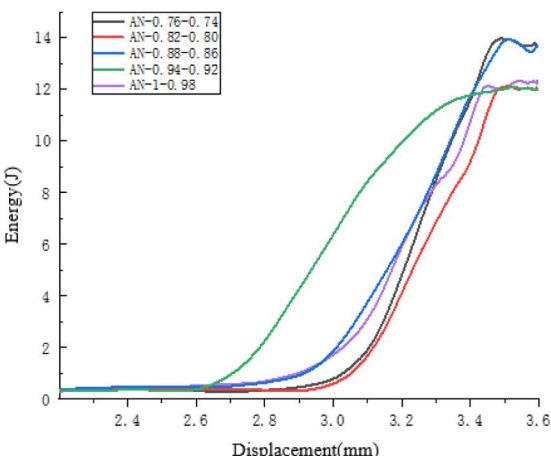
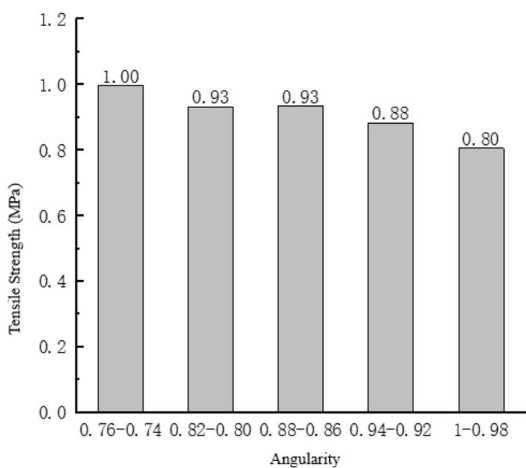

**Fig 26. The influence of angularity on the splitting resistance performance of the specimens.**

## 5. Discussion

Compared with previous studies, this research has achieved new progress in multiple aspects. Most previous studies mainly focused on the impact of a single aggregate morphological parameter on the performance of asphalt mixtures. However, through observing the morphology of recycled coarse aggregates, this study considered multiple parameters simultaneously and analyzed the relationships among them. This has enabled us to have a more comprehensive and in depth understanding of the relationship between aggregate morphology and mechanical properties. For example, some early studies [30–33] only focused on the aspect ratio or roundness ratio of aggregates, without considering the combined effects of shape parameters and angularity. This study found that the interactions among these parameters have a significant impact on the mechanical properties of the mixture and cannot be ignored.

In addition, the finite – element modeling algorithm adopted in this study provides a new method for researching the influence of aggregate morphology on the mechanical properties of asphalt mixtures. Compared with traditional research methods [34,35], finite element modeling can more intuitively display the stress distribution and deformation of specimens, providing more detailed information for the research. This is of great significance for deeply understanding the influence mechanism of aggregate morphology on mechanical properties.

## 6. Conclusions

This study primarily aimed to investigate the relationship between the distribution morphology of coarse aggregates and the anti-splitting performance of asphalt mixtures. By focusing on two microscopic characteristics of coarse aggregates, aspect ratio and shape parameters, and utilizing digital image processing technology and finite element software, the research concludes the following:

(1) Image processing technology is adopted to quantify the morphological characteristics of coarse aggregates. The aspect ratio and shape parameters are selected to evaluate the coarse aggregates of different particle size grades. Based on the random placement of aggregates, the Python language, which is a built-in script of the ABAQUS finite element software, is used to implement the random aggregate placement algorithm for asphalt mixtures, and numerical specimens are generated. The results combined with indoor tests show that this model can effectively reflect the comprehensive failure capacity of asphalt mixtures.

(2) For AC-16 asphalt mixtures, as the characteristic parameters change, the shapes of the aggregates become more and more irregular, and the overall compressive capacity shows an upward trend. Among them, the aspect ratio and angularity have the greatest influence on the compressive capacity of the specimens. Therefore, when the ratio of the long and short axes is too large, due to the significant difference in the horizontal and vertical mechanical capabilities of the aggregates, the overall compressive capacity of the specimens will decrease.

(3) In the splitting test, as the irregularity of the aggregates increases, the overall anti-splitting capacity of the specimens shows an upward trend. However, within the range of the shape parameter from 0.75 to 0.70 and the roundness ratio from 0.84 to 0.82, the anti-splitting capacity of the specimens decreases. That is to say, these two parameters impose certain limitations on the concavity and convexity of the aggregates, thereby reducing the anti-splitting performance of the asphalt mixtures.

## Supporting information

**S1 Appendix. The supporting information for this paper has been uploaded as an attachment, and the title of the file is Appendix.docx.**
(DOCX)

## Author contributions

**Conceptualization:** Nan Ru.

**Data curation:** Shuangchen Xia.

**Formal analysis:** Shuangchen Xia.

**Funding acquisition:** Shuangchen Xia.

**Investigation:** Nan Ru.

**Methodology:** Zhaoting Liu.

**Project administration:** Zhaoting Liu.

**Resources:** Nan Ru.

**Software:** Zhaoting Liu.

**Supervision:** Wencheng Cheng.

**Validation:** Nan Ru.

**Visualization:** Ya Li.

**Writing – original draft:** Wencheng Cheng.

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
