## [Decision Letter · Decision Letter 0]

PONE-D-24-55338

Research on the Influence of the Coarse Aggregates’ Morphology on the Splitting Performance of Asphalt Mixtures Based on Finite Element Modeling

PLOS ONE

Dear Dr. Ru,

Thank you for submitting your manuscript to PLOS ONE. After careful consideration, we feel that it has merit but does not fully meet PLOS ONE’s publication criteria as it currently stands. Therefore, we invite you to submit a revised version of the manuscript that addresses the points raised during the review process.

Thank you for submitting your article to PLOS ONE. The peer review process for your manuscript has been completed. Based on the reviewer’s comments, and upon my review of the manuscript, I feel that your paper presents information of interest to the readers of Plos One but that some revisions are needed prior to publication. Please see the comments of the reviewers on this manuscript. I suggest you consider these comments, suggestions, and questions and revise your article accordingly.

We look forward to receiving your revised manuscript.

Kind regards,

Ugur Ulusoy, Ph.D.

Academic Editor

PLOS ONE

Journal Requirements:

3. We note that your Data Availability Statement is currently as follows: “All relevant data are within the manuscript and in Supporting Information files.”

Reviewers' comments:

Reviewer's Responses to Questions

**Comments to the Author**

1. Is the manuscript technically sound, and do the data support the conclusions?

Reviewer #1: Partly

Reviewer #2: Yes

Reviewer #3: Yes

2. Has the statistical analysis been performed appropriately and rigorously? 

Reviewer #1: No

Reviewer #2: Yes

Reviewer #3: Yes

3. Have the authors made all data underlying the findings in their manuscript fully available?

Reviewer #1: No

Reviewer #2: Yes

Reviewer #3: Yes

4. Is the manuscript presented in an intelligible fashion and written in standard English?

Reviewer #1: Yes

Reviewer #2: Yes

Reviewer #3: Yes

5. Review Comments to the Author

Reviewer #1: Many key research details in this paper have not been discussed, and the conclusions are currently known and lack innovation.

1.The coarse aggregates morphological parameters are too simple to distinguish different coarse aggregates effectively.

2. The measurement process of coarse aggregate morphological parameters should be explained in detail.

3. Section 2.1, the steps and parameters for establishing the2D Digital Specimens should be detailed.

4. Section 3, few quantitative and in-depth research conclusions.

5. Section 4�conclusions are currently known and lack innovation.

Reviewer #2: In this paper, through finite element modeling, the morphology of coarse aggregate is effectively reflected, and the influence of aggregate morphology on the cracking resistance of asphalt mixture is accurately characterized. Generally, this paper needs some minor revisions before it is accepted for publication.

Reviewer #3: Complete review of English – adjust verb tenses.

Correct the font numbering in the text, placing it in ascending order.

The introduction must include the most current references, preferably from the last 5 years.

The article validates the accuracy of a two-dimensional finite element microscopic asphalt concrete model through an indoor Marshall splitting test. However, relying solely on this single test method may not provide a comprehensive overview, and it is advised to incorporate additional mechanical property tests.

The research primarily utilizes a two-dimensional model, which showcases the mechanical properties of asphalt mixtures to some extent. Nevertheless, there remain discrepancies when compared to the real three-dimensional structure. Establishing a three-dimensional digital specimen model is recommended to more accurately simulate the internal structure and mechanical behavior of asphalt mixtures.

While the effects of selected aspect ratio and shape parameters on splitting performance have been analyzed, there has been insufficient exploration of the sensitivity of these parameters. Furthermore, other morphological characteristics of aggregates could also influence the performance of asphalt mixtures. It is proposed to introduce additional parameters related to the morphological characteristics of aggregates to perform a more comprehensive analysis of their synergistic effects on asphalt mixture performance.

6. PLOS authors have the option to publish the peer review history of their article (what does this mean? ). If published, this will include your full peer review and any attached files.

**Do you want your identity to be public for this peer review?** For information about this choice, including consent withdrawal, please see our Privacy Policy .

Reviewer #1: No

Reviewer #2: No

Reviewer #3: No

---

## [Author Response · Author response to Decision Letter 1]

13 Mar 2025

Dear editors and review team:

I am extremely grateful to your journal for giving me this precious opportunity to revise and resubmit my manuscript. I also appreciate the time and effort spent by the review experts, who provided numerous highly constructive and guiding suggestions. During this revision process, I have conducted in-depth and meticulous research and responses to each and every suggestion from each reviewer.

For this revision, I have revisited a large number of authoritative literature sources to provide more detailed and accurate explanations of relevant concepts. I have also supplemented crucial experimental details to ensure the scientific nature and repeatability of the experiment. Meanwhile, I have reorganized the logical chain, presented the data in more intuitive chart forms, and accompanied them with in-depth textual analysis to enhance the persuasiveness of the results.

I sincerely hope that through this painstaking revision, the article can meet the academic standards of your journal and contribute a valuable reference to research in related fields. Thank you again for your hard work. I look forward to receiving further feedback.

---

## [Decision Letter · Decision Letter 1]

Research on the Influence of the Morphology of Coarse Aggregates from Reclaimed Asphalt Pavement (RAP) on the Mechanical Properties of Asphalt Concrete Based on Finite Element Modeling

PONE-D-24-55338R1

Dear Dr. Ru,

We’re pleased to inform you that your manuscript has been judged scientifically suitable for publication and will be formally accepted for publication once it meets all outstanding technical requirements.

Kind regards,

Ugur Ulusoy, Ph.D.

Academic Editor

PLOS ONE

Additional Editor Comments (optional):

Reviewers' comments:

Reviewer's Responses to Questions

**Comments to the Author**

1. If the authors have adequately addressed your comments raised in a previous round of review and you feel that this manuscript is now acceptable for publication, you may indicate that here to bypass the “Comments to the Author” section, enter your conflict of interest statement in the “Confidential to Editor” section, and submit your "Accept" recommendation.

Reviewer #3: All comments have been addressed

2. Is the manuscript technically sound, and do the data support the conclusions?

Reviewer #3: Yes

3. Has the statistical analysis been performed appropriately and rigorously? 

Reviewer #3: Yes

4. Have the authors made all data underlying the findings in their manuscript fully available?

Reviewer #3: Yes

5. Is the manuscript presented in an intelligible fashion and written in standard English?

Reviewer #3: Yes

6. Review Comments to the Author

Reviewer #3: Thank you, you made changes to the paper in response to all the questions I asked. I hope that the methods you proposed will be proven in practice.

7. PLOS authors have the option to publish the peer review history of their article (what does this mean? ). If published, this will include your full peer review and any attached files.

**Do you want your identity to be public for this peer review?** For information about this choice, including consent withdrawal, please see our Privacy Policy .

Reviewer #3: No

---

## [Editor Report · Acceptance letter]

PONE-D-24-55338R1

PLOS ONE

Dear Dr. Ru,

I'm pleased to inform you that your manuscript has been deemed suitable for publication in PLOS ONE. Congratulations! Your manuscript is now being handed over to our production team.

Kind regards,

on behalf of

Prof. Dr. Ugur Ulusoy

Academic Editor

PLOS ONE